# Antioxidant, Collagenase Inhibitory, and Antibacterial Effects of Bioactive Peptides Derived from Enzymatic Hydrolysate of *Ulva australis*

**DOI:** 10.3390/md21090469

**Published:** 2023-08-26

**Authors:** You-An Kang, Ye-Jin Kim, Sang-Keun Jin, Hwa-Jung Choi

**Affiliations:** 1Korea Beauty Industry Development Institute Co., Ltd., #501, Elite Bldg, Jeju Science Park, Cheomdanro 213-4, Jeju 63309, Republic of Korea; kangyouan@kbidi.or.kr; 2Oceanpep Co., Ltd., 105, Jinju Bioindustry Foundation, Musan-myeon, Jinju 52839, Republic of Korea; sssj4933@naver.com; 3Division of Animal Science, Gyeongsang National University, Jinju 52828, Republic of Korea; skjin@gnu.ac.kr; 4Department of Beauty Art, Youngsan University, 142 Bansong Beltway (Bansong-dong), Busan 48015, Republic of Korea

**Keywords:** multi-functional properties, *U. australis* hydrolysate, peptide, molecular docking

## Abstract

The protein extract of *Ulva australis* hydrolyzed with Alcalase and Flavourzyme was found to have multi-functional properties, including total antioxidant capacity (TAC), collagenase inhibitory, and antibacterial activities. The #5 fraction (SP5) and #7 fraction (SP7) of *U. australis* hydrolysate from cation-exchange chromatography displayed significantly high TAC, collagenase inhibitory, and antibacterial effects against *Propionibacterium acnes*, and only the Q3 fraction from anion-exchange chromatography displayed high multi-functional activities. Eight of 42 peptides identified by MALDI-TOF/MS and Q-TOF/MS/MS were selected from the results for screening with molecular docking on target proteins and were then synthesized. Thr-Gly-Thr-Trp (TGTW) displayed ABTS [2,2′-azino-bis (3-ethylbenzothiazoline-6-sulfonic acid)] radical scavenging activity. The effect of TAC as Trolox equivalence was dependent on the concentration of TGTW. Asn-Arg-Asp-Tyr (NRDY) and Arg-Asp-Arg-Phe (RDRF) exhibited collagenase inhibitory activity, which increased according to the increase in concentration, and their IC_50_ values were 0.95 mM and 0.84 mM, respectively. Peptides RDRF and His-Ala-Val-Tyr (HAVY) displayed anti-*P. Acnes* effects, with IC_50_ values of 8.57 mM and 13.23 mM, respectively. These results suggest that the *U. australis* hydrolysate could be a resource for the application of effective nutraceuticals and cosmetics.

## 1. Introduction

The genus *Ulva* (Ulvophyceae, Chlorophyta), consisting of common green macroalgae that often form algal beds, is found in intertidal zones in bays [1]. *U. australis* is commonly known as sea lettuce, an important marine vegetable and feed, which can be directly collected and eaten. *U. australis* has a high economic value for aquaculture [2]. *U. australis* is widely distributed on the coast of Korea and, particularly, it has appeared on the coast of Jeju island in amounts above 10,000 tons per year. Therefore, we need to use *U. australis* to create valuable products.

Although red and green algae contain 9–47% protein, most studies focus on a nutritional evaluation of total algae because of the difficulty of extracting cytoplasmic compounds from algae [3,4,5]. Recently, bioactive peptides from hydrolysate of seaweed, including antioxidant, anti-inflammatory, antimicrobial, and antihypertension, were widely investigated [6,7]. The peptides from the protein of macroalgae have potential applications to provide health benefits and treat disease [8,9,10,11]. The peptides from *Ulva* spp. were supposed to exhibit bioactivity, including the inhibition of angiotensin-I-converting enzyme (ACE), dipeptidyl peptidase IV, dipeptidyl peptidase III, renin, and α-glucosidase. They also showed antioxidant and anti-inflammatory effects via in silico analysis [12]. The papain hydrolysate of *U. lactuca* showed ACE inhibitory activity [13]. The peptides from gastro-intestinal digestion of *Ulva* spp. showed anti-inflammatory activity on immune cells [14].

Thanks to significant interest in therapeutic peptides, several protein–peptide docking techniques have been developed, leading the study and optimization of drug screening and design [15]. Protein–peptide interactions serve as structural components in approximately 40% of all macromolecular interactions [16,17]. Peptides can be used to prevent diseases involving malfunctioning of proteins due to undesirable protein–protein interactions [18]. Many databases and algorithms specifically in the field of peptide-based therapeutics have been developed in the past [19,20]. There are more than 200 therapeutic peptides approved by the FDA for the treatment of various diseases [21,22]. The aim of this research was to screen the multi-functional bioactivities of protein hydrolysate from *U. australis*. The protein from *U. australis* was hydrolyzed by Alcalase and Flavourzyme, the peptides from hydrolysate were purified by column chromatography, and the sequences of peptides were identified by MALDT-TOF and Q-TOF mass spectrometry. The in vitro ABTS radical scavenging effect, total antioxidant capacity, collagenase inhibitory, and anti-bacterial activities of peptides were evaluated after screening peptides through molecular docking.

## 2. Results

### 2.1. Properties of Hydrolysate from U. australis

The total amino acids of the powder of dried *U. australis* and the hydrolysate powder from dried *U. australis* were 17.17 g/100 g and 20.52 g/100 g, respectively, which increased to 3.35 g after hydrolysis (Table 1). Amino acids, including aspartic acid, glutamic acid, and proline were increased, while alanine and arginine were decreased after hydrolysis. The total amino acids of the powder of dried *U. australis* were high, ascending in the order of glutamic acid, alanine, and aspartic acid. For the identification of the health effects of non-proteinaceous amino acids such as taurine, we analyzed the free amino acid composition of hydrolysate from dried *U. australis*. The free amino acid content of hydrolysate from *U. australis* was 1296.3 mg/100 g, ascending in the order of threonine, glutamine, and arginine. Taurine, β-alanine, ornithine, and citrulline were identified as non-proteinaceous amino acids, but their contents were low (Table 1). It is suggested that ornithine and citrulline lead to the presence of a specific pathway for nitrogen metabolism in seaweed [23].

Solubility is an important factor in determining the reasonable sample amount for the application of chromatography. Solubilities of hydrolysate at 25 mM sodium citrate (pH 5.5) and 25 mM Tris-Cl (pH 8.0) were significantly decreased with the increase of the hydrolysate concentration by 40 mg/mL hydrolysate (*p* < 0.05), but were not the difference between pH 5.5 and pH 8.0 at 50 mg/mL. About 30% of hydrolysate was soluble at 50 mg/mL (Figure 1a). The solubility on pH of *U. australis* hydrolysate did not change in the range of pH 5.0~12 significantly, but greatly decreased below pH 4.5 (Figure 1b, *p* < 0.05 and *p* < 0.01). 

The hydrolysate from *U. australis* and the fractions of cation- and anion-exchange chromatography did not show the toxicities on RAW 264.7 cell (Figure 2).

### 2.2. Purification of Bioactive Peptides

The *U. australis* hydrolysate was separated into two fractions using a SP-Sepharose ion-exchange column, while separated into six fractions using a Q-Sepharose ion-exchange column (Figure 3). SP5 and SP7 were obtained from SP-Sepharose ion-exchange column chromatography. Q3, Q4, Q15, Q16, Q17, and Q23 were obtained from Q-Sepharose ion-exchange column chromatography. Relatively high TAC was observed in the three fractions: SP5, SP7, and Q3 (Table 2). A typical chromatogram of the purification procedure is shown in Figure 3. Total antioxidant activities of these three fractions were 1177.5, 674.1, and 489.1 μM of Trolox equivalent/mg-protein, respectively. Collagenase inhibitory and antibacterial against *P. Acnes* were also observed in the three fractions: SP5, SP7, and Q3 (Table 2). The three active fractions were divided further using size-exclusion chromatography. The SP 5 and SP7 fractions were divided into three fractions (S9, S19, and S21), while the Q3 fractions were into two fractions (S20 and S21). Bioactivities of fractions from size exclusion chromatography were not assayed further due to low protein concentration. The fractions from size exclusion chromatography were used for Q-TOF/MS/MS after complete dryness using a speed centrifuge at 40℃. The SP7/S9 fraction was purified on reversed phase (RP) chromatography more for MALDI-TOF analysis and divided into three fractions (SP7/S9/C4; SP7/S9/C5; SP7/S9/C20), while SP5/S9 fraction was not purified on RP chromatography because peptide peaks were not observed in MALDI-TOF analysis. The specific peptide spectrum of the SP5/S9 fraction was also not observed in Q-TOF/MS. The depression of NO production and cox-2 inhibitory activities were not observed in any fractions.

### 2.3. Amino Acid Sequence of Bioactive Peptides from Hydrolysate

To identify the sequence of peptides, the fractions (SP7/S9, SP7/S9, SP7/S19, and SP7/S21) were subjected to MALDI-TOF/MS and Q-TOF/MS/MS. Five peptides were identified in MALDI-TOF/MS, while 35 peptides were identified in Q-TOF/MS/MS. (Table 3). In MALDI-TOF/MS, the amino acid sequences from the protein database of *Ulva australis* (www.uniprot.org; 12 December 2022) were cleaved according to the substrate specificity of Alcalase proposed by Doucet et al. (2003) [24]. Database suggested the five protein sequences, including ferritin, ribulose biphosphate carboxylase-large chain, lectin, plastocyanin and chlorophyll apoprotein A1, derived from *Ulva australis*. The peptide sequence was identified from the molecular weight of cleaved fragments of protein sequences matched with the *m*/*z* value of the spectrum obtained from MALDI-TOF/MS. The sequences of 14 peptide sequences obtained from Q-TOF/MS/MS were identified from the database, but 21 peptide sequences were not identified. A typical mass spectrum of MALDI-TOF and Q-TOF/MS/MS is shown in Figure 4. The spectrum of SP7/S21 fraction using Q-TOF/MS/MS proposed the presence of glycan and suggested the possibility of *N*-acetylhexsosamine attached to serine and threonine residues of the protein. Since the protein from the database has 9–60 serine residues and 4–41 threonine residues, glycopeptide is possible in *U. australis* hydrolysate.

### 2.4. Molecular Docking for Screening Bioactive Peptides

In order to screen in silico bioactive peptides, eighteen peptides (Table 4) were docked into the active sites of target proteins, including 6WXV (pdb code) for antioxidant, 1HFC (pdb code) for collagenase inhibition, and 3NX7 (pdb code), using CDOCKER ligand docking module of BIOVIA Discovery Studio (Dassault Systems, Walsham, MA, USA). Protein, 6WXV, catalyzes the production of hydrogen peroxide as the FAD oxidase family, FAD and FDAH, is an important coenzyme for the mechanism of antioxidants. Protein, 1HFC, is collagenase in human fibroblast cells. Protein, 3NX7, is human matrix metallopeptidase-12 (MMP-12) [25]. The mechanisms of non-membrane targeting for antibacterial effect are dividing into an inhibition of protein and nucleic acid synthesis, an inhibition of serine protease, and a cell division [26]. Especially, eNAP-2 and indolcidine as antimicrobial peptides inhibited the elastase and chymotrypsin of microorganisms [27]. We supposed that inhibition of the selected protein for molecular docking leads to screening the bioactive peptides.

The reasonable docked conformations were selected based on cluster analysis and the Gold fitness score. FDPL, NRDY, and TGTW for antioxidant, NRDY and RDRF for collagenase inhibitor, and FDPL, HAVY, NRDY and RDRF for antibacterial effect were screened for in vitro validation based on molecular docking. The cluster number from 50 trials was 13 for TGTW as an antioxidant, 29 and 26 for NRDY and RDRF as a collagenase inhibitor, and 37 and 23 for HAVY and RDRF as an antibacterial effect, respectively. The Gold scores of peptides were in the range of 83.3–100.5 (Table 4).

Non-covalent interactions play an important role in bio-macromolecules, not only maintaining the three-dimensional structure of large molecules but also being responsible for the molecular recognition process [28]. The interaction models between the target proteins and peptides were analyzed by Discovery Studio (Figure 5). In the 6WXV/TGTW complex, TGTW has a hydrogen bond with Arg 1106, Met 1104, Tyr 1232, Arg 1214, and FAD 1601 within 6WXV, and a hydrophobic interaction with Ile 1100, Met 1205, His 1319, Trp 1222. The electrostatic interactions were formed with FAD 1601, Glu 1317, and Tyr 1228. In 1HFC/peptides, NRDY did not show an electrostatic interaction with Zn-ion in collagenase, while showed Glu 219, His 218, and His 222. In most cases of collagenase inhibition, inhibitory peptides formed an electrostatic interaction with Zn-ion. Although NRDY did not have an electrostatic interaction, the reason for having the collagenase inhibitory comparable to RDRF is ambiguous. HAVY as the antibacterial agent had a lot of hydrogen bonds compared to RDRF and showed a higher antibacterial effect than RDRF in vitro validation.

### 2.5. In Vitro Validation of Bioactive Peptides

ABTS [2,2′-azino-bis (3-ethylbenzothiazoline-6-sulfonic acid] scavenging and total antioxidant capacity of TGTW were dependent on the concentration. EC_50_ value of ABTS scavenging capacity was 0.14 mM comparable to 0.15 mM of ascorbic acid (Figure 6a). Total antioxidant capacity was 50 μM Trolox equivalent at the concentration of 53.9 μM (Figure 6b). The results suggest that *U. australis* containing TGTW is an available antioxidant agent for food application. NRDY and RDRF inhibited the collagenase in a concentration-dependent manner, the IC_50_ values of them were 0.95 mM and 0.84 mM, respectively (Figure 6c). Peptide, VICE, from the peptic hydrolysate of flounder skin inhibited collagenase up to 83.25% at 2.16 mM [29]. The collagenase inhibitory activity of RDRF was about 1.5 times higher than that of VICE. The survival of *P. Acnes* was greatly decreased with increasing the concentration of peptides, RDRF and HAVY, and the IC_50_ values were 8.57 mM and 13.23 mM, respectively (Figure 6d). The inhibition effect of RDRF was comparable with salicylic acid as positive control (8.35 mM).

Antimicrobial peptides are mainly composed of 12–100 amino acid residues, and their total charge of them is in the range of 2^+^–10^+^ as amphoteric peptides [30]. However, a long-chain peptide is not transported into the cell without a transporter. Therefore, we limited the amino acid residue of peptides below five amino acids in this study. Huan et al. (2020) classified the antimicrobial peptides based on amino acid-rich species as follows: proline-rich, tryptophan and arginine-rich, histidine-rich and glycine-rich peptides [31]. The stage of the action mechanism of small cationic peptides with antimicrobial activity is ruled by electrostatic interaction between the peptide and the pathogen cell membrane. An increase in its activity could be expected with an increase in the positive charge on the peptide [27]. Most cationic antimicrobial peptides act by accumulating on the surface of bacterial membranes and causing the formation of defects when a threshold is reached [32]. RDRF and HAVY had two positive charges, and the number of total positive charges was not much compared to other antimicrobial peptides. The peptide, SFIQRFTT, had two positive charges and suppressed the growth of *Listeria innocus* and *E. coli* [33].

## 3. Discussion

Many research efforts have focused on identifying effective peptides possessing biofunctional properties that might have preventive or therapeutic value to combat aging or disease. Protein hydrolysates are of interest as materials for medicinal purposes because of their pharmaceutical activity as inhibitors to reduce other related diseases and aging [34]. Some peptides having a potential angiotensin I-converting enzyme-inhibitory activity are inactive within the protein sequence, but they can reveal a biological activity when they are released by hydrolysis [35]. Moreover, among the various activities of protein hydrolysates, the inhibition of tyrosinase and elastase attracted attention due to their pharmaceutical activities in the fields of agriculture, food, and cosmetics as well as medicine [36].

In our study, the total and free amino acid composition of extracts and enzymatic hydrolysate were analyzed. As a result, acid amino acids, including aspartic acid, glutamic acid, and proline were increased, while alanine and arginine were decreased after hydrolysis. The total amino acid of dried *U. australis* was high in order to glutamic acid, alanine, and aspartic acid. Glutamic acid, proline, glycine, alanine and amino sulfonic acid are abundant in green algae [37]. Glutamic acid, aspartic acid, and leucine is abundant in *Enteromprpha prolifera* [38]. Our results were comparable to the amino acid composition of the previous reports. Furthermore, free amino acid content from *U. australis* hydrolysate was 1296.3 mg/100 g, high in order to threonine, glutamine, and arginine. The nitrogen compounds of the extract were composed of amino acids, peptides, and ammonia in the range of 40–70% [39]. Free amino acids from *Enteromorpha prolifera* depended on a harvested month, and arginine and glutamic acid were higher than other amino acids [40]. Green algae accumulate taurine and citrulline to regulate the osmotic pressure of seawater. Arginylglutamine as dipeptide is in *U. australis* and *U. linza* and is a major compound of free amino acid [41]. A lot of taurine and sarcosine were detected, while a little homocysteine and hydrolxylysine were also detected [40].

Solubility is an important factor in determining the reasonable sample amount for the application of chromatography. A previous study reported that the oyster hydrolysate did not show the pH-dependence [42]. In this study, the solubility on pH of *U. australis* hydrolysate did not change significantly in the range of pH 5.0~12, and solubilities of hydrolysate at 25 mM sodium citrate (pH 5.5) and 25 mM Tris-Cl (pH 8.0) were decreased with the increase of concentration, and there was not a difference of solubility between pH 5.5 and pH 8.0 at 50 mg/mL.

Numerous docking methods have been developed in the past for the structural determination of protein–peptide complexes and these methods can be classified broadly into the following three categories: (i) protein–peptide docking, (ii) protein-protein docking, and (iii) protein-small molecule docking [43]. Protein–peptide docking methods have been specifically developed to dock peptides on a protein [44]. Though protein–protein docking methods have been developed for docking two proteins, some of the software developed for docking small molecules on a protein can also be used to dock peptides on a protein [45]. Therefore, a wide range of docking methods can be used directly or indirectly for docking peptides on a protein. In this study, ABTS radical scavenging effect, TAC, collagenase inhibitory and anti-bacterial activities from enzymetic hydrolysate of *U. pertusa* were determined by protein–peptide docking techniques for the development of value-added products such as ingredients in foods or cosmetics. Eighteen peptides were docked into the active sites of target proteins. FDPL, NRDY, and TGTW for antioxidant, NRDY and RDRF for collagenase inhibitor, and FDPL, HAVY, NRDY and RDRF for antibacterial effect were selected based on cluster analysis and the Gold fitness score. Finally, TGTW, NRDY, NRDY and HAVY were confirmed by the interaction models of protein/peptide complex after molecular dynamics simulation by Discovery Studio. The final selected peptides showed ABTS radical scavenging effect (TGTW), total antioxidant capacity (TGTW), collagenase inhibitory (NRDY and RDRF) and antibacterial effect against *P. Acnes* (RDRF and HAVY). Overall, *U. australis* hydrolysate showed multi-function including ABTS radical scavenging effect, TAC, collagenase inhibition and antibacterial effect against *P. Acnes* and it can be used in food or cosmetic industries as a bioactive ingredient.

## 4. Materials and Methods

### 4.1. Materials

*U. australis* was collected from the coast of Jeju island, Korea in December 2021 and stored in a freezer at −20 °C after pulverizing dried *U. ustralis* under sunlight for 1 week. Alcalase 2.4 L (2.4 AU/g, endopeptidase, *Bacillus licheniformis*) and Flavourzyme 500 MG (500 LAPG/g, endoprotease and exopeptidase, *Aspergillus oryzae*) were purchased from Biosis company (Busan, Korea). Water and acetonitrile (AH365-4, Honeywell, Burdick & Jackson Co., MI, USA) were high-performance liquid chromatography (HPLC) grade. The mouse macrophage cell line RAW 264.7 was obtained from the Korean Cell Line Bank (Seoul, Korea). Dulbecco’s Modified Eagle’s Medium (DMEM) supplemented with 10% fetal bovine serum (FBS) and 1% penicillin–streptomycin was supplied by Gibco BRL (Grand Island, NY, USA). Assay kits for total antioxidant capacity (ab65329) and collagenase inhibitory activity (EnzChek gelatinase/collagenase) were purchased from Abcam Inc. (Cambridge, UK) and Invitrogen Co. (Waltham, MA, USA), respectively. HiLoad SP- and Q-Sepharose, and HiLoad 16/60 Superdex 30 prep grade columns were purchased from GE Healthcare (Parsoppany, NJ, USA). Eight peptides (FDPL, TGTW, HAVY, HVIA, NRDY, LPYPG, PETF and RDRF) were synthesized at A&PEP Co. (Cheongju, Korea). The purity and molecular weight were 97.2% and 490.54 Da for FDPL; 97.0% and 463.48 Da for TGTW; 95.1% and 488.53 Da for HAVY; 95.3% and 438.52 Da for HVIA; 97.6% and 566.56 Da for NRDY; 97.5% and 545.62 Da for LPYPG; 97.4% and 492.52 Da for PETF; and 97.0% and 592.64 Da, respectively.

### 4.2. Preparation of Hydrolysate from U. australis

Extract of protein from *U. australis* was conducted by alkaline method [46]. Dried *U. australis* (100 g) mixed with 15 volumes of 1N NaOH for 4 h and alkali-soluble protein was recovered as supernatants obtained by centrifuge (Supra R22, Hanil Sci. Co., Ltd., Dajeon, Korea) at 8000 rpm for 10 min. After adjustment to pH 4.2 using 1N HCl, the precipitated protein was collected by centrifuge at 8000 rpm for 10 min. After suspending with 3 volumes of distilled water, the suspension was adjusted to pH 7.0. One percentage (*w*/*w*) of Alcalase 2.4 L and Flavourzyme 500 MG were added to the suspension of precipitated protein, and the mixture was then incubated at 60 °C for 4 h with stirring at 60 rpm. After inactivating the protease in a 95–100 °C water bath for 1 h, the hydrolysate was centrifuged at 8000 rpm for 15 min, and the supernatant was lyophilized and stored at −20 °C.

### 4.3. Solubility of Hydrolysate

To analyze solubility according to pH, distilled water was added into enzymatic hydrolysate (0.2 g) and adjusted at 20 mL. Each 2 mL from them adjusted into pH 1–13 ranges with 1N HCL or 1N NaOH. The protein concentration of supernatant by centrifuging (Labogen 1248, Gyrozen Co., Ltd., Gimpo, Korea; 2500× *g* and 10 min) was measured with the Biuret method [22]. The relative solubility was expressed as a percentage of soluble proteins/sample proteins.

### 4.4. Analysis of Total and Free Amino Acid Composition

Sun dried powder of *U. australis* (10 mg) and enzymatic hydrolysate (10 mg) were weighed exactly in a test tube. After adding 1.5 mL of 6 N HCL, the test tube was filled with N2 gas and sealed. Acid hydrolysis was conducted using a heating block (HB-96D, Daihan Scientific Co., Ltd., Seoul, Korea) for 24 h. The hydrolysate was filtered with a 3G-4 glass filter, and the hydrochloride of the filtrate was completely evaporated with a vacuum rotatory evaporator (N-1110, Eyela, Tokyo, Japan) below 50 °C. The acid hydrolysate was dissolved in 0.02 N HCl solution, and filtered with a 0.02 µm cylinder-type filter. After injecting 40 μL of filtrate, the amino acid composition was analyzed by the Amino Acid Analyzer (Biochrom 30, Biochrom, Cambridge, UK). The total amino acid was analyzed with a sodium buffer system, while the free amino acid composition was analyzed with a lithium buffer system.

### 4.5. Purification of Bioactive Peptides from Hydrolysate

*U. australis* hydrolysates were dissolved in 25 mM citrate buffer (pH 5.5) for cation-exchanger chromatography and 25 mM Tris-Cl buffer (pH 7.5) for anion-exchanger chromatography, respectively, and the bioactive peptide fractions were eluted using a HiLoad SP-Sepharose column (16 × 100 mm) and HiLoad Q-Sepharose column (16 × 100 mm). Elution was performed using a linear gradient system from solvent A (25 mM citrate, pH 5.5) to solvent B (25 mM citrate, pH 5.5 containing 0.6 M NaCl) for cation-exchanger chromatography, and from solvent A (25 mM Tris-Cl, Ph 7.5) to solvent B (25 mM Tris-Cl, pH 7.5 containing 0.6 M NaCl) for anion-exchange chromatography with 10 column volumes at a flow rate of 4 mL/min, and detected at 220 nm and 280 nm. Antioxidant effect, collagenase inhibition, NO production, COX-2 inhibition and antibacterial effect against *P. Acnes* were assayed for each peak. The bioactive fractions were pooled and concentrated with a vacuum rotary evaporator. The concentrated fractions were loaded on a HiLoad Superdex 30 prep grade column (16 × 600 mm) and eluted with HPLC grade water at a flow of 1 mL/min. Detection was then carried out at 220 nm and 280 nm. Each fraction was lyophilized and used for Q-TOF/MS/MS analysis. For MALDI-TOF analysis, each fraction from size-exclusion chromatography was dissolved in 0.1% TFA/water and fractionated using the UHPLC system (Ultimate 3000, Thermo Fisher Scientific, Walsham, MA, USA) with a Biobasic C18 column (4.6 × 250 mm, 5 µm; Thermo Fisher Scientific, Walsham, MA, USA). Elution was performed using a linear gradient system from solvent A (0.1% TFA/water) to solvent B (0.1% TFA/50% ACN) over 10 min at a flow rate of 1 mL/min at 40 °C and detected at 220 nm. The column was equilibrated with solvent A, after which 50 μL of sample was applied to the column and elution was carried as follows: 10 min of solvent A, 2 min of A-B gradient, 10 min of solvent B, and 10 min of solvent A for equilibration. Each fraction was dried in a speed vacuum concentrator and determined the *m*/*z* value in MALDI-TOF spectrometry.

### 4.6. Cell Toxicity of the Hydrolysate

The cytotoxicity of hydrolysate and each fraction was measured at murine macrophage cell line RAW 264.7 (KCLB, Seoul, Korea) by the MTS assay described previously [41]. Cell viability is calculated by the following equation: cell viability (%) = (1 − At/Ac) × 100, where At is the sample treated absorbance and Ac is the absorbance of the non-treated control.

### 4.7. ABTS Radical Scavenging Activity and Total Antioxidant Capacity (TAC)

ABTS radical scavenging activity was determined by a modification of a previously reported method [26]. ABTS was calculated by the following equation: ABTS (%) = (Ac − At/Ac) × 100, where At is the sample treated absorbance and Ac is the absorbance of the non-treated control. TAC (%) was expressed as Trolox equivalent capacity (μM).

### 4.8. Inhibitory Activity of NO Production

Marine macrophage cell line RAW 264.7 (KCLB, Seoul, Republic of Korea) was cultured in Dulbecco’s Modified Eagle’s Medium (DMEM) supplemented with 10% fetal bovine serum (FBS) and 1% penicillin–streptomycin (Gibco, NY, USA) at 37 °C in a humidified atmosphere containing 5% CO_2_. After subculture, RAW 264.7 cells were seeded in 24-well plates (5 × 10^5^ cells/well) in DMEM and incubated for 12 h. The adherent cells were treated with a test sample together with LPS (1 μg/mL) for 24 h. Following incubation for 24 h, the medium was collected and medium nitrate concentration was measured by Griess reagent as an indicator of NO production. A NaNO2 standard curve was used to calculate the nitrite concentration for control and samples. The NO-suppressing activity was expressed as the percent of the control value, which was the nitrite concentration from only LPS-activated RAW 264.7 cells.

### 4.9. COX-2 Inhibitory Activity

Cyclooxygenase 2 (COX-2) inhibitory activity was measured using a COX-2 inhibitor screening kit (Fluorometric K547-100, BioVision Inc., Milpitas, CA, USA) according to the manufacturer’s instructions. Fluorescent intensity was measured at excitation 535 nm and emission 580 nm for 10 min using an ELISA plate reader. COX-2 inhibitory activity was calculated by the following equation: COX-2 inhibitory activity (%) = 1 − (slope of sample/slope of control × 100).

### 4.10. Collagenase Inhibitory Activity

Collagenase inhibitory activity was determined using EnzChek^®^ Gelatinase/Collagenase Assay Kit (Invitrogen Therapeutcics Inc, Austin, TX, USA) [32]. Collagenase inhibitory activity (%) was calculated by the following equation: collagenase inhibitory activity (%) = [100 − (B − Bc/A − Ac)] × 100, where A is non-sample and collagenase-treated absorbance, B is sample and collagenase-treated absorbance, Ac is non-sample and non-collagenase-treated absorbance, and Bc is non-sample and collagenase-treated absorbance.

### 4.11. Antibacterial Effect on P. Acnes

The *P. Acnes* were cultivated in Reinforced Clostridial medium broth. The concentration of *P. Acnes* was adjusted to 0.6 OD 600 nm. The diluted microbial suspension (100 µL) and samples (40 µL) were inoculated into a 96-well microplate. After incubation for 1 day at 37 °C, the absorbance was measured at 600 nm with a microplate reader (VersaMax ELISA Microplate Reader, Molecular Devices, San Jose, CA. USA). The results were transformed into a percentage of the controls.

### 4.12. MALDI-TOF Mass and Q-TOF Mass/Mass Spectrometry

The fractions from size-exclusion were completely dried in a vacuum centrifuge. The dried samples were dissolved in TA30 (30:70 (*v*/*v*) acetonitril:0.1% trifluoroacetic acid in water) and mixed with a saturated solution of α-cyano-4-hydroxycinnamic acid. The mixture was spotted onto polished steel targets (Bruker Daltonics, Bremen, Germany). Mass spectra were acquired using an autoflex MAX MALDI-TOF/TOF (Bruker Daltonics; High-tech Materials Analysis Core Facility at Gyeongsang National University, Jinju, Republic of Korea), equipped with Smartbeam II laser working at 355 nm. The instrument was operated in the positive ion reflection mode. An external calibration was performed using a Peptide Calibration Standard (Bruker Daltonics), which includes angiotensin II (MH+ = 1046.5418), angiotensin I (MH+ = 1296.6848), substance P (MH+ = 1347.7354), bombesin (MH+ = 1619.8223), ACTH clip 1–17 (MH+ = 2093.0862), ACTH clip 18–39 (MH+ = 2465.1983), and somatostatin 28 (MH+ = 3147.4710). Mass spectra in the range of 300–4000 Da were obtained by averaging 500 laser shots and peptide peaks were generated using FlexAnalysis software ver 3.4 (Bruker Daltonics). The MS/MS spectra were obtained in the LIFT mode. De novo peptide sequencing of the major peaks was performed by manual interpretation of the spectra compared with the sequence obtained from the *Ulva* protein database (www.uniprot.org; 12 December 2022).

UHPLC-Q-TOF/MS/MS analyses were performed using a Vanquish UHPLC (Thermo Fisher Scientific, Walsham, MA, USA) coupled to a Q Extractive plus (Thermo Fisher Scientific, Walsham, MA, USA) mass spectrum spectrometer equipped with an electrospray ionization (ESI) source. Each lyophilized fraction from size-exclusion chromatography was dissolved in 200 μL of 0.1% formic acid and filtered with 0.20 µm membrane filter after centrifuge at 13,000 rpm for 10 min.

Chromatographic separation was carried out on Zobax eclipse plus C18 column (3.0 × 100 mm, 1.8 µm, Agilent). The mobile phase consisted of solvent A (0.1% formic acid/water, *v*/*v*) and solvent B (0.1% formic acid/acetonitrile, *v*/*v*) and elution conditions were as follows: 0.0–2.0, 2% B; 2.0–30.0 min, 30% B; 30.0–31.0, 98% B; 31.0–36.0, 98% B; 36.0–37.0, 2.0% B; 37–45.0 min, and 2% B. The flow rate was set at 0.3 mL/min. The column and autosampler were maintained at 30 °C and 4 °C, respectively. The injection volume was 5 μL. Mass spectrometry was set to acquire over the *m*/*z* range 100–1500 with full MS scan type. For the electrospray ion source, spray voltage and capillary temperature were set to 3.5 kV and 250 °C, respectively. MS/MS parameter was set to Dd-MS2(HCD) as scan type, 17,500 as resolution, 100 ms as maximum ion time and 100 *m*/*z* as fixed first mass. Both the peptide sequencing module of the software and manual calculations were used to process the MS/MS data and to perform peptide sequencing.

### 4.13. Molecular Docking

The calculation of target protein–peptide docking was performed with genetic optimization (GOLD version 5.2.2, Cambridge Crystallographic Data Centre, Cambridge, UK) according to a previous method [47]. Synthesized peptides (HVIA, TGTW, HAVY, NRDY, RDRF, PETF, RDRF and FDPL) were used as the ligand, and the 3D structures were obtained from the Discovery Studio (DS) program (BIOVIA, San Diego, CA, USA). Geometry was optimized by energy minimizing using minimized ligand tool existing DS. Ligand converted to SD file for use of GOLD and contained all active ligands within a 10 Å radius of the center for calculation. The consistency of the 3D structure in each peptide was identified by clustering 50 times. The mouse DUOX1 (pdb, 6WXV), human fibroblast collagenase (pdb, 1HFC), and human MMP-12 (pdb, 3NX7) as targets for protein–peptide docking were downloaded from www.rcsb.org (12 April 2023).

### 4.14. Statistical Analysis

Data were expressed as the mean with standard deviation of triplicate determinations. Analysis of variance was carried out by the Tukey HSD test using the JMP (version 12, SAS Institute, Cary, NC, USA). Significance was indicated at *p* < 0.05 and *p* < 0.01.

## 5. Conclusions

The fractions from *U. australis* hydrolysate using cation-exchange chromatography showed high ABTS scavenging and TAC, the inhibition of collagenase, and an antibacterial effect against *P. Acnes*. *U. australis* hydrolysate and the fractions from ion-exchange chromatography did not show toxicities on RAW 264.7 cells. In TGTW, the ABTS radical scavenging capacity was comparable to ascorbic acid, and the total antioxidant capacity of 50 uM Trolox was equivalent at the concentration of 53.9 μM. The IC_50_ values of NRDY and RDRF for collagenase inhibition were 0.95 mM and 0.84 mM, respectively. The antibacterial effect of RDRF against *P. Acnes* showed IC_50_ values of 8.57 mM compared with salicylic acid. Since the *U. australis* hydrolysate showed multi-function including antioxidant, collagenase inhibition, and antibacterial effect against *P. Acnes*, it has a potential for cosmetic application.

## Figures and Tables

**Figure 1 marinedrugs-21-00469-f001:**
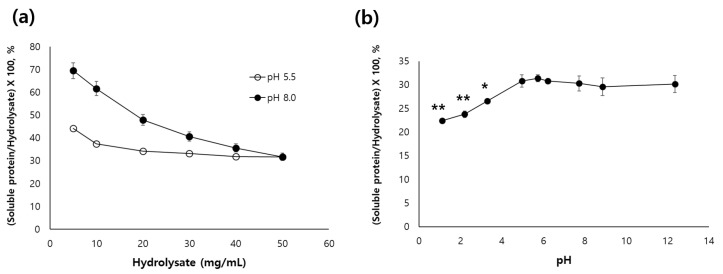
Effect of hydrolysate concentration (**a**) and pH (**b**) on the solubility. * Significantly different at *p* < 0.05. ** Significantly different at *p* < 0.01.

**Figure 2 marinedrugs-21-00469-f002:**
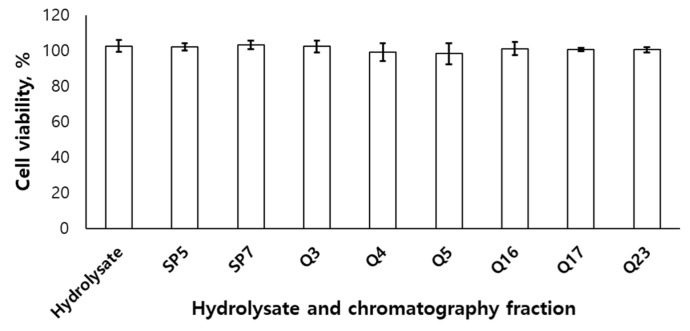
Viability of RAW 264.7 cell line of hydrolysate and ion-exchange fractions from Ulva australis hydrolysate.

**Figure 3 marinedrugs-21-00469-f003:**
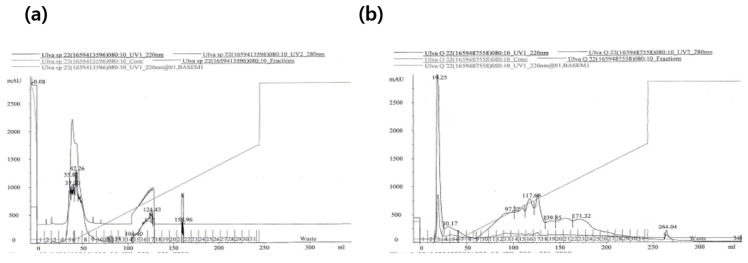
Separation procedure of bioactive peptide by cation- (**a**) and anion- (**b**) exchange chromatography of the hydrolysate of *U. australis*. Cation-exchange chromatography was performed with a linear gradient of 25 mM citrate containing 0.6 M NaCl, pH 5.5 on a HiLoad SP-Sepharose column (16 × 100 mm), and anion-exchange chromatography was performed with a linear gradient of 25 mM Tris-Cl containing 0.6 M NaCl, pH 8.0 on a HiLoad Q-Sepharose column (16 × 100 mm). The flow rate was 4 mL/min. The eluent was detected at 280 nm and 220 nm using a UV detector and fractioned with 5 mL. The number below and above of x-axis represents the fraction volume, and the fraction number.

**Figure 4 marinedrugs-21-00469-f004:**
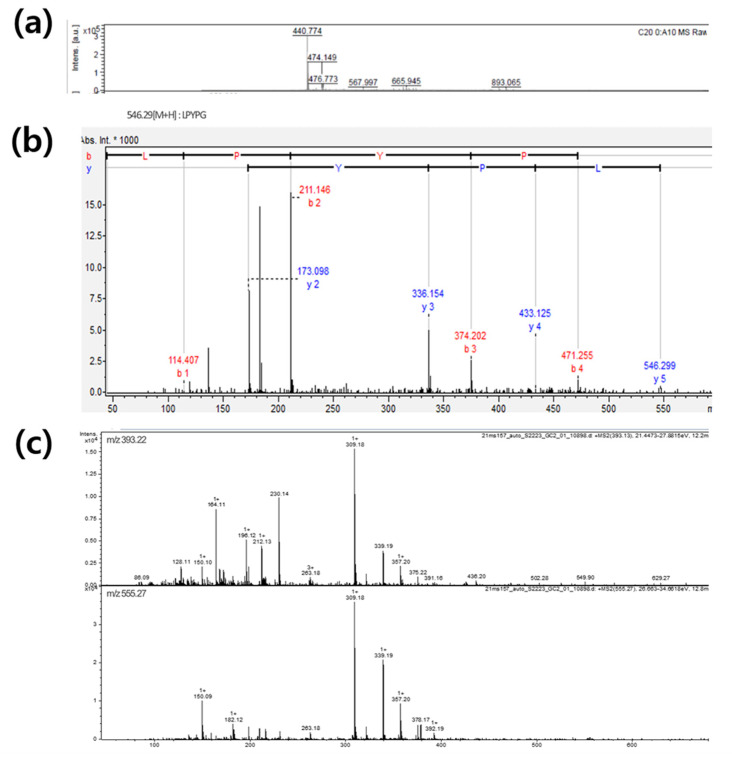
A typical mass spectrum of MALDI-TOF of NRDY (**a**), Q-TOF/MS/MS of LPYPG (**b**), and Q-TOF/MS/MS product ions of compounds presumed to be glycopeptide (**c**) for identification of peptides sequence and glycopeptide from the hydrolysate of *Ulva australis*.

**Figure 5 marinedrugs-21-00469-f005:**
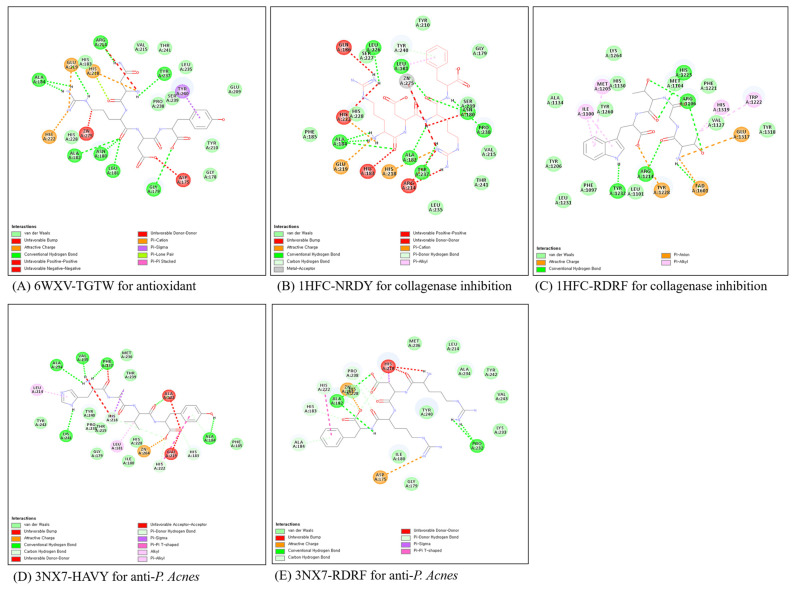
The interaction models of protein/peptide complex after molecular dynamics simulation by Discovery Studio. Number represents the order number of amino acid residue of protein sequence. Green, blue and yellow represent vander Walls interaction, conventional hydrogen bond and attractive charge.

**Figure 6 marinedrugs-21-00469-f006:**
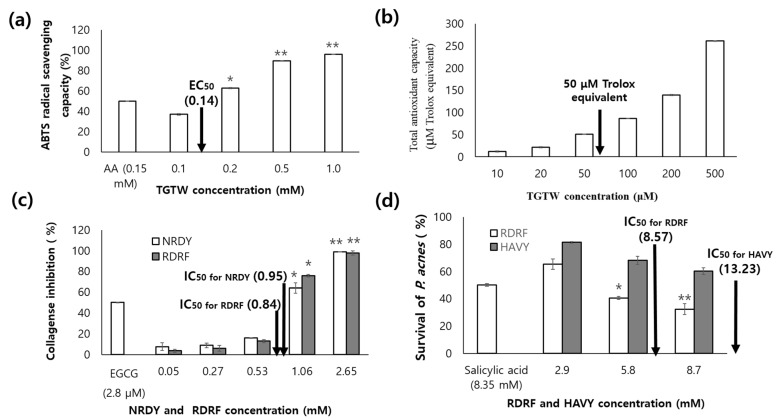
ABTS radical scavenging effect (**a**); total antioxidant capacity (**b**); collagenase inhibitory (**c**); and antibacterial effect against *P. Acnes* (**d**) of the synthetic peptides. * Significantly different at *p* < 0.05. ** Significantly different at *p* < 0.01. AA in (**a**) represents ascorbic acid. EGCG in (**b**) represents epigallocatechin gallate.

**Table 1 marinedrugs-21-00469-t001:** Total amino acid and free amino acid composition of the powder of dried *Ulva australis* and its hydrolysate.

Amino Acid	Total Amino Acid ^1^ (g/100 g)		Free Amino Acid ^2^ (mg/100 g-Hydrolysate)
	Powder of Dried *U. australis*	Hydrolysate	
Asp	1.70 ± 0.14	2.47 ± 0.03	63.6 ± 0.6
Thr	0.88 ± 0.08	1.03 ± 0.02	285.6 ± 3.8
Ser	1.08 ± 0.00	0.92 ± 0.03	45.7 ± 0.8
Glu	2.31 ± 0.09	3.03 ± 0.04	147.3 ± 4.6
Pro	0.57 ± 0.08	1.06 ± 0.01	35.9 ± 0.8
Gly	1.31 ± 0.09	1.14 ± 0.08	56.6 ± 0.4
Ala	1.86 ± 0.00	1.57 ± 0.07	63.6 ± 0.3
Cys	0.09 ± 0.01	0.14 ± 0.01	21.0 ± 1.1
Val	1.07 ± 0.09	1.31 ± 0.03	30.5 ± 0.3
Met	0.34 ± 0.02	0.53 ± 0.03	22.6 ± 6.6
Ile	0.74 ± 0.04	0.87 ± 0.03	9.0 ± 1.8
Leu	1.36 ± 0.06	1.56 ± 0.03	81.2 ± 4.2
Tyr	0.59 ± 0.05	1.20 ± 0.08	42.1 ± 9.2
Phe	0.95 ± 0.04	1.27 ± 0.08	33.3 ± 9.9
His	0.10 ± 0.00	0.14 ± 0.02	0
Lys	0.59 ± 0.01	0.62 ± 0.04	32.8 ± 0.2
NH3	0.31 ± 0.02	0.31 ± 0.01	17.9 ± 0.1
Arg	1.30 ± 0.04	1.06 ± 0.05	113.3 ± 0.8
Taurine	ND ^3^	ND	40.6 ± 0.5
Sarcosine	ND	ND	15.4 ± 1.6
Citrulline	ND	ND	15.4 ± 0.2
α-amino butyric acid	ND	ND	5.0 ± 0.1
Cystathionine	ND	ND	7.3 ± 2.7
β-Alanine	ND	ND	38.1 ± 14.7
Homocysteine	ND	ND	17.7 ± 3.4
Hydroxylysine	ND	ND	12.8 ± 0.4
Ornithine	ND	ND	32.8 ± 0.2
Total	17.16 ± 0.05	20.52 ± 1.01	1296.3 ± 60.2

^1^ The powder of dried samples was hydrolyzed by 6 HCl, and analyzed with sodium buffer system in amino acid analyzer. ^2^ Hydrolysate dissolved in water was precipitated with 5′-salfosalycylic acid, and removed by centrifugation. The supernatants were analyzed with lithium buffer system in amino acid analyzer. ^3^ ND represents non-detection in sodium buffer system.

**Table 2 marinedrugs-21-00469-t002:** Total antioxidant capacity (TAC), NO production, COX-2 inhibition, collagenase inhibition, and anti-bacterial activity against *P. Acnes* of the fraction from ion-exchange chromatography.

Fraction		Protein (mg/mL)	TAC (μM Trolox Equivalent/ mg-Protein)	NO Production (%)	COX-2 Inhibition (%)	Collagenase Inhibition (%)	Survival of *P. Acnes* (%)
Hydrolysate		14.85	382.0 ± 0.0	115.4 ± 3.5	66.0 ± 0.6	87.1 ± 7.4	112.0 ± 0.3
Cation-Chrom.	SP5	1.57	1177.5 ± 29.3 *	127.4 ± 2.5 *	0	27.4 ± 0.3	19.1 ± 1.6 *
	SP7	2.81	674.1 ± 7.8 *	130.8 ± 3.4 *	0	4.2 ± 1.4	18.6 ± 1.1 *
Anion-Chrom.	Q3	6.76	489.1 ± 10.8 *	119.8 ± 4.5	11.2 ± 3.2	98.9 ± 0.1 *	5.5 ± 0.3 *
	Q4	6.74	24.7 ± 8.0	99.3 ± 5.1 *	17.3 ± 0.9	0	123.7 ± 14.8
	Q15	6.6	56.5 ± 1.2	124.2 ± 2.3 *	0	0	98.2 ± 6.7
	Q16	5.92	33.6 ± 0.3	118.9 ± 5.4	4.3 ± 3.2	0	97.8 ± 2.6
	Q17	6.6	38.3 ± 0.6	125.2 ± 7.0	0	0	102.0 ± 4.8
	Q23	7.02	13.4± 0.5	127.1 ± 8.0	18.1 ± 2.4	0	56.2 ± 4.3 *

SP and Q represent cation and anion resins for ion-exchange chromatography, respectively. Number represents the pooled fraction number. Values are expressed as means ± standard deviation from three independent experiments (*n* = 5). * Significantly different from hydrolysate (*p* < 0.05). TAC, total antioxidant capacity.

**Table 3 marinedrugs-21-00469-t003:** The sequence of the peptide fragments identified from *Ulva australis* hydrolysate using MALDI-TOF/MS and Micro Q-TOF/MS/MS spectrometer.

MS	Fraction ^(1)^	RT (min)	z	*m*/*z*	Molecular Weight, Da	Sequence	Database
MALDI-TOF/MS	SP7/S9			439.818	438.52	HVIA	lectin
				464.863	463.48	TGTW	ribulose
				488.901	488.53	HAVY	ferritin
				567.912	566.56	NRDY	lectin
				593.005	592.64	RDRF	ribulose
Micro Q-TOF III MS/MS	SP7/S9	11.64	1	231.18	230.17	VL	ribulose
		14.89	1	245.18	244.18	LL	ferritin ribulose apoprotein A1
	SP7/S19-20	14	1	445.23	444.23	DPTL	ND
		15.7	1	493.23	492.23	PETF	ND
		17.1	1	546.29	545.29	LPYPG	ND
		17.4	1	675.33	674.33	LPYPGE	ND
		18.2	1	578.31	577.31	FTPLT	ND
		18.5	1	489.27	488.27	LPYP	ND
		20.3	1	344.24	343.24	LVL	ND
		20.4	1	399.26	398.26	PLGL	ND
		20.9	1	491.25	490.25	FDPL	ND
		24.6	1	560.34	559.34	PALLF	ND
	SP7/S21	13.1	1	338.17	337.17	VGY	ND
		17.1	1	546.29	545.29	LPYPG	ND
		17.3	1	463.25	462.25	IAYP	ribulose
		17.4	1	352.19	351.19	VSF	ND
		20.9	1	378.24	377.24	MF	apoprotein A1
	Q3/S20	2.11	1	265.123	264.123	VF	ferritin ribulose lectin plastocyanin apoprotein A1
		3.35	1	233.15	232.15	LT	ribulose apoprotein A1
		3.9	1	279.138	278.138	LF	ribulose
		5.57	1	217.156	216.156	VV	ribulose lectin plastocyanin apoprotein A1
		6.92	1	228.193	227.193	VGL	ferrtin
		7.6	1	274.177	273.177	ALA	lectin apoprotein A1
		8.02	1	286.177	287.177	PLG	ribulose
		14.34	1	318.204	317.204	SVL	ND
		14.39	2	265.19	528.38	AVKVL	ND
		17.8	1	316.224	315.224	ALL	ND
		19.77	2	286.664	571.328	GVHLF	ND
		20.02	1	344.256	343.256	VLL	ND
		20.96	1	502.326	501.326	LSGLL	ND
		21.06	1	399.261	398.261	PLGL	ND
		22.61	1	502.326	501.326	LGVTL	ND
	Q3/S21	7.05	1	203.14	202.14	AL	lectin
		9.26	1	231.171	230.171	LV	apoprotein A1
		9.59	1	229.156	228.156	PL	ferritin

^(1)^ SP and Q represent cation and anion resins for ion-exchange chromatography, respectively. Numbers represent fractions obtained from each chromatogram. S represents a size exclusion chromatography. Molecular weight comes from ChemSpider (www.chemspider.com; 21 August 2023). ND means there is no protein database (www.uniprot.org; 12 December 2022) of *Ulva australis*.

**Table 4 marinedrugs-21-00469-t004:** Cluster number and Gold score of the synthetic peptides against their target proteins.

Peptide/Target Protein	Antioxidant Effect		Collagenase Inhibition		Antibacterial Effect	
	6WXV		1HFC		3NX7	
	Cluster	Gold Score	Cluster	Gold Score	Cluster	Gold Score
DPTL	14	70.4	12	60.2	50	84
FDPL	17	85.4	8	57.3	39	86.9
FTPLT	14	82.1	16	60.8	14	80.5
HAVY	9	79.5	32	68.5	37	93.8
HVIA	27	76.6	29	62.4	46	85.3
IAYP	9	79.6	28	65.7	17	81
LPYP	7	74.3	8	55.4	14	59.3
LPYPG	4	71.2	21	58.4	6	61.3
LVL	31	68.6	16	60.5	18	76.2
MF	9	65.6	38	55	30	70.5
NRDY	8	100.5	29	85.9	14	97.1
PALLF	4	85.5	15	67.8	7	76.3
PETE	7	87.4	36	67.4	28	83
PLGL	8	68.9	44	67.8	48	76.8
RDRF	3	101.3	26	83.3	23	99.7
TGTW	13	89.7	27	67.4	16	85.3
VGY	11	54.1	42	64	50	77.8
VSF	29	70	23	63.1	39	80.4

## Data Availability

All the data generated or analyzed during this study were included in this article.

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
