# Peer review of "Antioxidant, Collagenase Inhibitory, and Antibacterial Effects of Bioactive Peptides Derived from Enzymatic Hydrolysate of Ulva australis"

_marinedrugs, 2023, doi:10.3390/md21090469_

Round 1

Reviewer 1 Report

The manuscript fits into the research area that Marine Drugs deals with and meets the general standards for publication. Then, I think the paper can be acceptable for publication after a minor revision.

1. All the indexes of Table 1 have not been fully discussed, especially the free amino acid components that the hydrolysate does not contain. Please confirm wether sample means dry powder of U. australis.

2. Two statements of line82- line84 contradict each other. Significant characters are not stated in Figure 1.

3. More latest references should be cited.

Author Response

Response to Reviewers

Reviewer 1

Comments and Suggestions for Authors

The manuscript fits into the research area that Marine Drugs deals with and meets the general standards for publication. Then, I think the paper can be acceptable for publication after a minor revision.

  1. All the indexes of Table 1 have not been fully discussed, especially the free amino acid components that the hydrolysate does not contain. Please confirm wether sample means dry powder of australis.

>Thanks, we try to explain the sample mean and free amino acid component. We improve on our weakness, and insert in the Table and text.

The samples used as powder of dried conditions. Thus modified into “powder of dried sample” at Table 1 and text. Also, free amino acid only measured at U. australis hydroysate .

>Unify the sample condition as “powder of a dried sample” and “hydrolysate” in the Table 1 and text. Since the objectives of this study is to screen the bioactive of peptides from the hydrolysate from U. australis, we didn’t analyze the free amino acid composition of the powder of a dried sample.

  1. Two statements of line82- line84 contradict each other. Significant characters are not stated in Figure 1.

>Figure 1a indicates the solubility of hydrolysate dissolved in pH 5.5 and 8.0 according to the amount of hydrolysate, whereas figure 1b indicates the effect of pH on the solubility of hydrolysate at 50 mg/mL. The figure 1b showed the 30% solubilities pH 5.5 and pH 8.0. The r results of the hydrolysate at 50 mg/mL at pH 5.5 and 8.0(Fig.1a) were equal to the results of pH dependence of Fig1b at 50 mg/mL. I think that the results not contradict each other. The sentence was corrected for clear explanation. The significance can be verified by the standard deviation in figures.

>Figure 1a and Figure 1b changed. Significant characters of p<0.01 inserted at legend of Figure 2 and stated at text.

Fig.1a and Fig1b were not changed. Fig 1a showed the effect of hydrolysate concentration on the solubility, while Fig.1b showed the effect of pH on solubility at 50 mg/mL. We stated the significant characts in the text.

  1. More latest references should be cited.

More latest references cited and “Pyeun, J.H.; Jeon, J.G. Seafood chemistry, Suhaksa, Seoul, South Korea., 1994; pp.283–355.” inserted.

Reviewer 2 Report

This manuscript described the potential bioactivities of peptides derived from Ulva australis hydrolysate. Total antioxidant capacity (TAC), inhibition of collagenase, and antibacterial properties were observed from several fractions like #5 (SP5) and #7 (SP7). MALDI-TOF/MS and Q-TOF/MS/MS analyses revealed 42 promising peptides, and eight were chosen for molecular docking on target proteins and subsequently synthesized. Some of them displayed bioactivities with IC50 values of 0.84~13.23 mM. Overall, this manuscript is poorly organized and written. The reviewer does not recommend the publication on Marine Drugs due to the following concerns.

1)     Language should be improved.

2)     About the identification of the peptide fragments using MALDI-TOF/MS and Micro Q-TOF/MS/MS spectrometer, it looks like the differences between calculated MW and the experimental values are significant.

3)     The resulting potency is too weak. If the authors are willing to develop nutraceuticals and cosmetics, more appropriate journals are supposed to choose. For drug discovery, these peptides are not good candidates.

4)     For the synthesized peptides, more evidences should be provided to show their purity and validity.

Many grammar mistakes and format issues.

Author Response

Response to Reviewers

Reviewer 2

Comments and Suggestions for Authors

This manuscript described the potential bioactivities of peptides derived from Ulva australis hydrolysate. Total antioxidant capacity (TAC), inhibition of collagenase, and antibacterial properties were observed from several fractions like #5 (SP5) and #7 (SP7). MALDI-TOF/MS and Q-TOF/MS/MS analyses revealed 42 promising peptides, and eight were chosen for molecular docking on target proteins and subsequently synthesized. Some of them displayed bioactivities with IC50 values of 0.84~13.23 mM. Overall, this manuscript is poorly organized and written. The reviewer does not recommend the publication on Marine Drugs due to the following concerns.

1)     Language should be improved.

> Language improved.

2)     About the identification of the peptide fragments using MALDI-TOF/MS and Micro Q-TOF/MS/MS spectrometer, it looks like the differences between calculated MW and the experimental values are significant.

> The sequence of peptide fragments was identified by m/z value of mass list of SP7/S9 fraction from MALDI-TOF/MS and the sequence data base of proteins in Ulva pertusa (protein entry name: Q6T6H8, BOYGZ5 and) H9KVI6 using de novo sequencing. The sequence of peptide fragments was analyzed in Protein Works (Dajeon, Korea) using UHPLC- Q-TOF/MS/MS spectrometer. We display the mass list of MALDI-TOF/MS as supplement-1. Table 3 modified.

Supplement-1: Mass list of SP7/S9 fraction

m/z               S/N       Quality Fac.   Res.            Intens.            Area

440.720        430       28516         4196            52744             7923

442.722        144       34867         4054            17679             2736

567.959        114       40201         4726            15507             2802

659.184        103       26048         5139            13966             2870

460.739         64       11084          4043             8008             1342

444.757         46       8533         3858             5766              952

476.725         45       4222         4305             5745              958

671.925         45       15773         4491             6123             1533

665.887         43       8439         4430             5834             1471

614.102         42       11106        4173             5745             1287

482.873         37       13049       3918             4717              852

523.930         35         9477       4036             4610              890

877.035         33         3595       4132             3850             1490

454.714         30         2328       4186             3709              620

892.999         30         1527       4340             3448             1351

655.949         29         7472       4800             3923              854

567.912         25       33818       1570             3825             2016

438.752         25          450       2516             3061              795

681.855         24         1202       4772             3247              837

687.896         23         3657       4620             3212              819

586.950         22         3443       4901             3023              611

649.919         20         4621       4343             2763              648

549.927         19         5164       4626             2586              470

456.863         19          766       4149             2448              399

861.075         19         1030       4472             2314              766

 675.180         17           1126      4271              2365               570

593.005         15           2476      2232              3691              1549

598.092         15           1056      3555              2048               505

519.903         15            500       3704              2051               463

637.171         13           1249      3873             1779               457

464.838         12            588        3818            1554              273

908.975         10           893             4427           1170              467

856.993          9           611             4874           1111               386

439.818          9           26.1            4492           1163              180

522.883          8           199             3311            1117              282

464.863          6           154             2573           1597              414

488.901          5           171             1196             792              452

3)     The resulting potency is too weak. If the authors are willing to develop nutraceuticals and cosmetics, more appropriate journals are supposed to choose. For drug discovery, these peptides are not good candidates.

>You are right. Macroalgae is not good resource for a functional activity of peptide. Most functional activity derived from macroalgae comes from polysaccharide, polyphenol, flavonoid and pigments etc. Protein content of Ulva spp. Represents between 5.8 and 24.8% of dry weight (Pyun and Jeon, 1994). However, macroalgae stand for a sustainable source of proteins that can be applied as ingredients for human and animal consumption (Echave et al., 2021: Marine drugs: doi.org/10.3390/md19090500). If new technologies are developed for pretreatment prior protein extraction, involving disruptive techniques of cell wall. Protein and hydrolysate from Ulva spp. are good source for human health.

Collagenase inhibitory activities of GPM from Alaska pollack skin, GPK from oyster hydrolysate, VDL and VICE from flounder skin were 3.92, 19.07, 13.46 and 83.35% at 1 mg/mL, respectively (Kim et al., 2022; J Korean Soc Food Sci Nutr; doi/org/10.3746/jkfn.2022.51.4.322). Therefore, IC 50 value (0.84 mM) of RDRF from Ulva australis for collagenase inhibition is lower than that of VICE from a flounder skin. IC50 value of RDRF against P. acnes was comparable to the that of salicylic acid. Salicylic acid of 0.5~1.0% was added to cleasing cream.

4)     For the synthesized peptides, more evidences should be provided to show their purity and validity.

>Purity and molecular weight of the synthetic peptides were inserted in Materials and display it as supplement-2.

Supplement-2. Product specification of the synthesized peptides

Product name

Peptide sequence

Formula

Molecular weight, Da

Qty, mg

Purity, %

Lot No.

FL-4

FDPL

C24H34N4O7

490.54

200.0

97.22

P220714-MX1004266

TW-4

TGTW

C21H29N5O7

463.48

200.0

97.07

P220701-MX1001219

HY-4

HAVY

C23H32N6O6

488.53

200.0

95.14

P220701-MX1001220

HA-4

HVIA

C20H34N6O5

438.52

200.0

95.39

P220701-MX1001221

NY-4

NRDY

C23H34N8O9

566.56

200.0

97.62

P220701-MX1001222

LG-5

LPYPG

C27H39N6O7

545.62

200.0

97.58

P220714-MX1004266

PF-4

PETF

C23H32N4O8

492.52

200.0

97.45

P220714-MX1004267

RF-4

RDRF

C26H40N10O27

592.64

200.0

97.08

P220714-MX1004268

Reviewer 3 Report

This manuscript uses Ulva australis as the raw material, starting from the functional requirements of cosmetics such as antioxidant, anti collagen degradation, and anti acne, to explore the active protein peptides, which has certain research and application value. The author separated and identified peptide sequences using anion/cation adsorption chromatography, exclusion chromatography, and HPLC, and validated the efficacy and activity of the active peptide using bioinformatics technology and in vitro activity analysis. The research of the article is in line with the scope of this journal, and has certain innovation and reference significance. But there are still some issues that need to be clarified before the article can be accepted, as follows:

1) line 105, Please provide exclusion chromatography and high-performance liquid chromatography separation chromatograms.

2) line 113, Although no inhibition of NO production and cox-2 inhibitory activity were observed, experimental methods and results need to be provided as evidence.

3) Table 2. Please indicate the pooled fraction numbers for each samples(SP5, SP7, Q3, Q4......).

4) line 156. Please revise the PDB protein name error and incomplete expression in the first sentence of section 2.4.

5) line 158. Remove the space symbol from the protein name "6 WXV" in line 158.

6) Table 4. Are reported active peptide sequences set as positive controls? Using commercial or literature reported active peptide sequences as positive controls will have more reference value.

7)line 304. Does the percentage data here refer to the dry weight of protein?

8) line 337. Lack of experimental methods for cell culture and detection of NO release.

9) line 394. Please provide the Denovo identification results of the peptide sequence to demonstrate the reliability of sequence identification.

The English quality of the manuscript is acceptable.

Author Response

Response to Reviewers

Reviewer 3

Comments and Suggestions for Authors

This manuscript uses Ulva australis as the raw material, starting from the functional requirements of cosmetics such as antioxidant, anti collagen degradation, and anti acne, to explore the active protein peptides, which has certain research and application value. The author separated and identified peptide sequences using anion/cation adsorption chromatography, exclusion chromatography, and HPLC, and validated the efficacy and activity of the active peptide using bioinformatics technology and in vitro activity analysis. The research of the article is in line with the scope of this journal, and has certain innovation and reference significance. But there are still some issues that need to be clarified before the article can be accepted, as follows:

1) line 105, Please provide exclusion chromatography and high-performance liquid chromatography separation chromatograms.

Line105: Size exclusion chromatogram didn’t contain in the text because of much figure, Preparative RP-HPLC was not performed because RP-HPLC is performed in the stage of Q-TOF/MS/MS.

>Our results contained many tables and figure. Therefore, this data displayed as supplement data. Size exclusion chromatogram is following as:

Supplement-3. Size exclusion chromatogram of SP 7 fraction, and the 9-10 fractions,19-20 fractions and 21 fraction were pooled for further analysis.

Supplement-4. Size exclusion chromatogram of Q3 fraction, and the 20 fraction and 21-22 fractions and were pooled for further analysis

2) line 113, Although no inhibition of NO production and cox-2 inhibitory activity were observed, experimental methods and results need to be provided as evidence.

>Experimental methods for NO production and COX-2 inhibitory activity were inserted in “Materials and Methods’ section. Also, results were inserted in Table 2.

3) Table 2. Please indicate the pooled fraction numbers for each samples(SP5, SP7, Q3, Q4......).

>“SP5 and SP7 obtained from SP-Sepharose ion-exchange column chromatography. Q3, Q4, Q15, Q16, Q17 and Q23 obtained from Q-Sepharose ion exchange column chromatography.” inserted at “2.2. purification of bioactive peptides”.

4) line 156. Please revise the PDB protein name error and incomplete expression in the first sentence of section 2.4.

>iHFC was revised into 1HFC. The first sentence also was revised into “In order to screen in silico bioactive peptides, eighteen peptides (Table 4) were docked into the active sites of target proteins, including 6WXV (pdb code) for antioxidant, 1HFC (pdb code) for collagenase inhibition, and 3NX7 (pdb code), using CDOCKER ligand docking module of BIOVIA Discovery Studio (Dassault Systems, Walsham, MA, USA).”

5) line 158. Remove the space symbol from the protein name "6 WXV" in line 158.

>Thank you for your attention. We removed the space of “6 WXV.

6) Table 4. Are reported active peptide sequences set as positive controls? Using commercial or literature reported active peptide sequences as positive controls will have more reference value.

>All peptides in Table 4 come from the hydrolysate of U. australis. Commercial and the active peptides reported in literature didn’t use.

7)line 304. Does the percentage data here refer to the dry weight of protein?

>One percentage represents the ratio of protease to the precipitated protein from dried U.australis. 4.2 section was revised. 

8) line 337. Lack of experimental methods for cell culture and detection of NO release.

>Experimental methods for cell culture and NO detection present in “4.8. Inhibitory activity of NO production”

9) line 394. Please provide the Denovo identification results of the peptide sequence to demonstrate the reliability of sequence identification.

>The sequence of peptide fragments was identified by comparison of m/z value of mass list of SP7/S9 fraction from MALDI-TOF/MS and the sequence data base of proteins in Ulva pertusa (protein entry name: Q6T6H8, BOYGZ5 and) H9KVI6. This data displayed as supplement data.

Supplement-1: Mass list of SP7/S9 fraction

m/z               S/N       Quality Fac.   Res.            Intens.            Area

440.720        430       28516         4196            52744             7923

442.722        144       34867         4054            17679             2736

567.959        114       40201         4726            15507             2802

659.184        103       26048         5139            13966             2870

460.739         64       11084          4043             8008             1342

444.757         46       8533         3858             5766              952

476.725         45       4222         4305             5745              958

671.925         45       15773         4491             6123             1533

665.887         43       8439         4430             5834             1471

614.102         42       11106        4173             5745             1287

482.873         37       13049       3918             4717              852

523.930         35         9477       4036             4610              890

877.035         33         3595       4132             3850             1490

454.714         30         2328       4186             3709              620

892.999         30         1527       4340             3448             1351

655.949         29        7472       4800             3923              854

567.912(NRDY) 25        33818       1570             3825             2016

438.752         25          450       2516             3061              795

681.855         24         1202       4772             3247              837

687.896         23         3657       4620             3212              819

586.950         22         3443       4901             3023              611

649.919         20         4621       4343             2763              648

549.927         19         5164       4626             2586              470

456.863         19          766       4149             2448              399

861.075         19         1030       4472             2314              766

675.180         17         1126       4271            2365              570

593.005(RDRF)  15         2476      2232            3691             1549

598.092         15          1056      3555            2048              505

519.903         15           500      3704            2051              463

637.171         13          1249      3873            1779              457

464.838         12           588      3818            1554              273

908.975         10           893      4427            1170              467

856.993          9           611      4874            1111               386

439.818 (HVIA)  9           26.1      4492           1163              180

522.883          8           199             3311            1117              282

464.863(TGTW)  6           154             2573           1597              414

488.901(HAVY)  5           171             1196             792              452

Amino acid sequence of protein from U. australis (www.uniprot.org)

>tr|H9KVI6|H9KVI6_ULVPE Ferritin OS=Ulva pertusa OX=3120 PE=1 SV=1

AQEVTGMVFQPFSEVQGELSTVTQAPVTDSYARVEYHIECEAAINEQINIEYTISYVYHA

LHSYFARDNVGLPGFAKFFKEASDEEREHAHMLMDYQTKRGGRVELKPLAAPEMEFANDD

KGEALYAMELALSLEKLNFQKLQALQAIADKHKDAALCDFVEGGLLSEQVDAVKEHAVYV

SQLRRVGKGVGVYLLDQELGEEEA

Biological process: Iron storage

Ligand: Iron, metal-binding

>tr|B0YGZ5|B0YGZ5_ULVPE Ribulose bisphosphate carboxylase large chain (Fragment) OS=Ulva pertusa OX=3120 GN=rbcL PE=3 SV=1AFRMTPQPGVPAEEAGAAVAAESSTGTWTTVWTDGLTSLDRYKGRCYDIEPLGEDDQYIAYIAYPLDLFEEGSVTNLFTSIVGNVFGFKALRALRLEDLRIPPAYVKTFQGPPHGIQVERDKLNKYGRGLLGCTIKPKLGLSAKNYGRAVYECLRGGLDFTKDDENVNSQPFMRWRDRFLFVAEAIYKSQSETGEVKGHYLNATAGTCEEMMERGQFAKDLGVPIVMHDYITGGFTANTSLAHFCRASGLLLHIHRAMHAVIDRQRNHGIHFRVLAKILRMSGGDHLHSGTVVGKLXGEREITL

Biological process: Calvin cycle, carbon dioxide fixation, photorespiration, photosynthesis

Ligand: magnesium, metal-binding

>sp|Q6T6H8|LEC_ULVPE Lectin OS=Ulva pertusa OX=3120 GN=UPL1 PE=1 SV=1MINILHVIAGLALASVGVDARQVGVGADVLHAVENTIDSITGVEASHSALEVGGGITNTDNWETFAGLPLTGAIKVNDGNSVVHISAYFPEDRRGKYSYYAATSDELQKTVVFLFVVEDDGLLLQAVKNNAHYPVTNGMYLASHRYYPKDSKYEGMVRLMVHADPAKAVIWEFVTVGGKQYLKVKENRDYTALQIPRHHPRPG

>sp|P56274|PLAS_ULVPE Plastocyanin OS=Ulva pertusa OX=3120 GN=PETE PE=1 SV=1AQIVKLGGDDGSLAFVPSKISVAAGEAIEFVNNAGFPHNIVFDEDAVPAGVDADAISYDDYLNSKGETVVRKLSTPGVYGVYCEPHAGAGMKMTITVQ

Biological process: antiviral defense, electron transport, transport

Ligand: copper, metal-binding

Round 2

Reviewer 2 Report

1. Please provide the characterization data (1H NMR, 13C NMR, etc) for the synthsized peptides.

2. Figure 5 is not clear. Please provide high-quality picture.

Fine.

Author Response

Response to Reviewers

Reviewer 2

Comments and Suggestions for Authors

  1. Please provide the characterization data (1H NMR, 13C NMR, etc) for the synthesized peptides.

>Synthetic peptides were composed of L-amino acids, and synthesized by solid phase of three step-partially protected amino acid, the formation of peptide bond, and cleavage of the protecting groups. Finally, the synthetic peptide was purified by RP-HPLC. Conventionally, The purity and identification of synthetic peptides were identified by HPLC and m/z values of mass spectrometry, respectively (Gross and Caprioli, 2005. The encyclopedia of mass spectrometry, Elsevier, NY, pp.165-176).

 In peptide research, NMR is usually used to verify specific interactions of the compound with protein and identification of the binding site on the protein (Moy et al., 2001. Analytical Chem, 73: 571-581).

We submit the example of peptide, RDRF, to verify the purity and m/z values of mass spectrum (Figure 1 and Figure 2).

Fig. 1. RP-HPLC chromatogram of peptide, RDRF, for identification of purity.

Fig. 2. Mass spectrum of peptide, RDRF, for identification of mass(m/z).

  1. Figure 5 is not clear. Please provide high-quality picture.

>Resolution of Figure 5 increased with 600 dpi to show high-quality picture. It submitted as figure file.
